# Experimental Study on Laser-MIG Hybrid Welding of Thick High-Mn Steel Plate for Cryogenic Tank Production

**Du-Song Kim [1,2], Hee-Keun Lee [2], Woo-Jae Seong [2], Kwang-Hyeon Lee [3] and Hee-Seon Bang [4],***

1 Department of Welding and Joining Science Engineering, Graduate School, Chosun University, 309 Pilmun-daero, Dong-Gu, Gwangju 501759, Korea; dusong@dsme.co.kr
2 Welding Engineering R&D Department, Daewoo Shipbuilding Marine Engineering, Geoje 53302, Korea; zetlee@dsme.co.kr (H.-K.L.); wjseong@dsme.co.kr (W.-J.S.)
3 Busan Machinery Research Center, Korea Institute of Machinery & Materials, Busan 46239, Korea; leekh@kimm.re.kr
4 Department of Welding and Joining Science Engineering, Chosun University, 309 Pilmun-daero, Dong-Gu, Gwangju 501759, Korea
* Correspondence: banghs@chosun.ac.kr; Tel.: +82-62-230-7215

**Abstract:** The International Maritime Organization has recently updated the ship emission standards to reduce atmospheric contamination. One technique for reducing emissions involves using liquefied natural gas (LNG). The tanks used for the transport and storage of LNG must have very low thermal expansion and high cryogenic toughness. For excellent cryogenic properties, high-Mn steel with a complete austenitic structure is used to design these tanks. We aim to determine the optimum welding conditions for performing Laser-MIG (Metal Inert Gas) hybrid welding through the MIG leading and laser following processes. A welding speed of 100 cm/min was used for welding a 15 mm thick high-Mn steel plate. The welding performance was evaluated through mechanical property tests (tensile and yield strength, low-temperature impact, hardness) of the welded joints after performing the experiment. As a result, it was confirmed that the tensile strength was slightly less than 818.4 MPa, and the yield strength was 30% higher than base material. The low-temperature impact values were equal to or greater than 58 J at all locations in the weld zone. The hardness test confirmed that the hardness did not exceed 292 HV. The results of this study indicate that it is possible to use laser-MIG hybrid welding on thick high-Mn steel plates.

**Keywords:** high-Mn steel; laser-MIG hybrid welding; cryogenic tank; humping defect; low-temperature impact test

## 1. Introduction

The International Maritime Organization (IMO) is encouraging the application of ships fueled by liquefied natural gas (LNG) to raise the emission standards for ships, thereby reducing the atmospheric contamination caused by them [1–3]. Since LNG is transported as a liquid, a cargo hold that can store and transport LNG at cryogenic temperatures of −163 °C is required [3]. However, at such low temperatures, the application of common steel, including SS400, has several constraints due to its brittleness. Therefore, it is necessary to use a metal for the construction of an LNG tank that does not become brittle at cryogenic temperatures. The IMO International Code for the Construction and Equipment of Ships Carrying Liquefied Gases in Bulk (IGC Code) specifies that 9% nickel steel with a high cryogenic impact toughness secured, stainless steel (e.g., A240-304L), 36% nickel steel (Invar), AL5083-O, and high-Mn steel are limited for use in LNG tank materials [4,5] (Table 1). The criteria for the mechanical strength of these cryogenic steels are defined according to the IGC Code.

**Table 1.** Materials for cryogenic fuels listed in the IGC Code.

| Material | Chemical Composition | Yield Strength (MPa) | Tensile Strength (MPa) |
|---|---|---|---|
| 9% Nickel steel | Fe-9Ni | >585 | 690–825 |
| STS304L | Fe-18.5Cr-9.25Ni | >205 | >585 |
| Al5083-O | Al-4.5Mg | 124–200 | 276–352 |
| Invar | Fe-36Ni | 230–350 | 400–500 |
| High-Mn steel | Medium C-high Mn | >400 | 800–970 |

The presence of Ni in most of these steels has increased the process costs due to the rising unit prices, which is attributed to the global rise in commercial volume and export restrictions [6]. As a result, high-Mn steel has been developed as an alternative material that displays excellent cryogenic properties by adding a large amount of relatively inexpensive Mn [7–9]. The LNG fuel tanks of ships are made of a minimum of 10 mm thick high-Mn steels depending on the design requirements. However, arc welding processes that are frequently used on ships and steel structures, such as flux-cored arc welding (FCAW) and submerged arc welding (SAW) [10,11], are difficult to use owing to their limited penetration depth. It is necessary to account for the productivity of the welding process using multilayer welding to perform welding on a thick plate. Although an increased heat input in arc welding can improve the productivity of the welding process, it may damage the properties of the welded joints and lead to unexpected distortion when exposed to the welding heat, which may reduce the total productivity.

Laser and laser-arc hybrid welding processes can replace such arc welding processes because they are capable of functioning at high welding speeds, reducing welding distortion, and increasing penetration depth [12]. The application of laser welding alone requires precise machining of the weld joints because the gap tolerance of the joint is determined by the size of the laser beam shortcomings, which were addressed by developing a laser-arc hybrid welding process that utilized laser and arc welding simultaneously.

The proposed welding process used the advantages of arc welding to generate a wide gap tolerance and the advantages of laser welding to provide deep penetration [13]. Further, it improved the welding productivity and quality.

Recently, various studies of cryogenic steels specified in Table 1 have been conducted. However, research on laser welding of high-Mn steel has been extremely limited. Kim et al. studied the effect of major laser welding parameters of penetration on 9% nickel steel. The mechanical properties were compared after performing laser welding and FCAW welding experiments [14,15]. Tayebi et al. compared the mechanical properties after performing tungsten inert gas (TIG) welding and Nd:YAG laser welding tests for STS304L and STS316 materials [16]. Kim et al. applied the cross-shaped structure of the Invar material to the LNG tank structure derived through the cold wire laser welding (CWLW) process [17]. Khan et al. studied the alloying element loss and microstructure deformation of high-Mn austenitic stainless steels according to the type of laser welding, welding speed, laser power, and shielding gas flow rate [18]. In a recent study that is closest to the purpose of this study, Kim et al. identified the relationship between welding parameters and penetration features by changing laser output and welding speed through bead-on-plate (BOP) test with a fiber laser welding process [19].

This study utilizes the laser-MIG hybrid welding process with optimal butt welding of a 15 mm thick high-Mn steel plate. The welding productivity was maximized by studying the ideal welding conditions with a minimum welding speed of 100 cm/min. An I-groove was made (without additional machining), and one-side welding for butt joint was welded.

The test piece was then evaluated under optimal welding conditions to verify whether it met the quality standards required by the IGC Code. Table 2 below summarizes the objectives of this study and the quality requirements of the IGC Code.

**Table 2.** Research objectives and IGC Code requirements.

| Key Performance Indicators | Units | Research Objectives | Remarks |
|---|---|---|---|
| Welding joint/groove | - | Butt joint/I-groove | - |
| Welding pass | Pass | One pass | - |
| Welding speed | cm/min | ≥100 | - |
| * Tensile strength | MPa | ≥660 | Base metal (min. T.S.: 800 MPa) or welding consumable (min. T.S.: 660 MPa), whichever is lower |
| * Impact energy | J | ≥27 J at −196 °C | At weld metal base |
| * Bend test | - | Defect is not acceptable | Defects appearing at the corners of a test specimen may be disregarded |
| * Macro test | - | Defect is not acceptable | Cracks and lack of fusion are not accepted |
| * Hardness test | - | Hardness values are only for information | - |

* IGC Code requirements.

## 2. Materials and Methods

### 2.1. Experimental Specifications and Welding Parameters

This study analyzes the optimal conditions for carrying out laser-MIG hybrid welding on 15 mm thick high-Mn steel specimens. The high-Mn steel used in this study contained more than 25% Mn. The chemical compositions of the base and welding consumables used are summarized in Table 3.

**Table 3.** Chemical compositions of high-Mn steel (base material) and the welding consumable.

| Material | Chemical Composition (%) | | | | |
|---|---|---|---|---|---|
| High-Mn steel (ASTM A240XM-M) | C | Si | Mn | P | S |
| | 0.4417 | 0.282 | 24.291 | 0.0155 | 0.0008 |
| | Cr | Ni | As | B (ppm) | - |
| | 3.380 | 0.027 | 0.010 | 28 | - |
| Welding consumable (ASME SFA-5.22 E307T1-1/4) | C | Si | Mn | P | S |
| | 0.302 | 0.423 | 19.56 | 0.008 | 0.001 |

The experimental specimens were fabricated according to the butt joint welding conditions of the outer shell of the LNG fuel tank, as shown in Figure 1. Two plates were constructed under conditions suitable for butt joint welding with an I-groove without additional machining. The dimensions of the test specimens were 400 mm × 200 mm × 15 mm. All specimens were tested continuously. No additional backing material was attached to the back side of the specimen. One-side welding for the butt joint was performed, and two laser-MIG hybrid welding systems were used. The laser-MIG hybrid welding process in this study was such that the MIG led, and the laser followed. The fundamental experiment was conducted using a 16 kW disk laser-MIG hybrid system with a 400 A arc welding power supply, a push–pull feeder, and a laser hybrid welding system combined with a 6-axis industrial high-precision robot. The main experiment was conducted using a high-power 20 kW fiber laser, a 400 A arc welding power source, a push–pull feeding system, and a 4-axis computer numerical control. The 16 kW disk laser and the 20 kW fiber laser were transmitted one to one through a 300 μm optical cable and

irradiated upon the specimen with identical laser head configurations. The minimum diameter of the laser beam was equal to 0.29 mm.

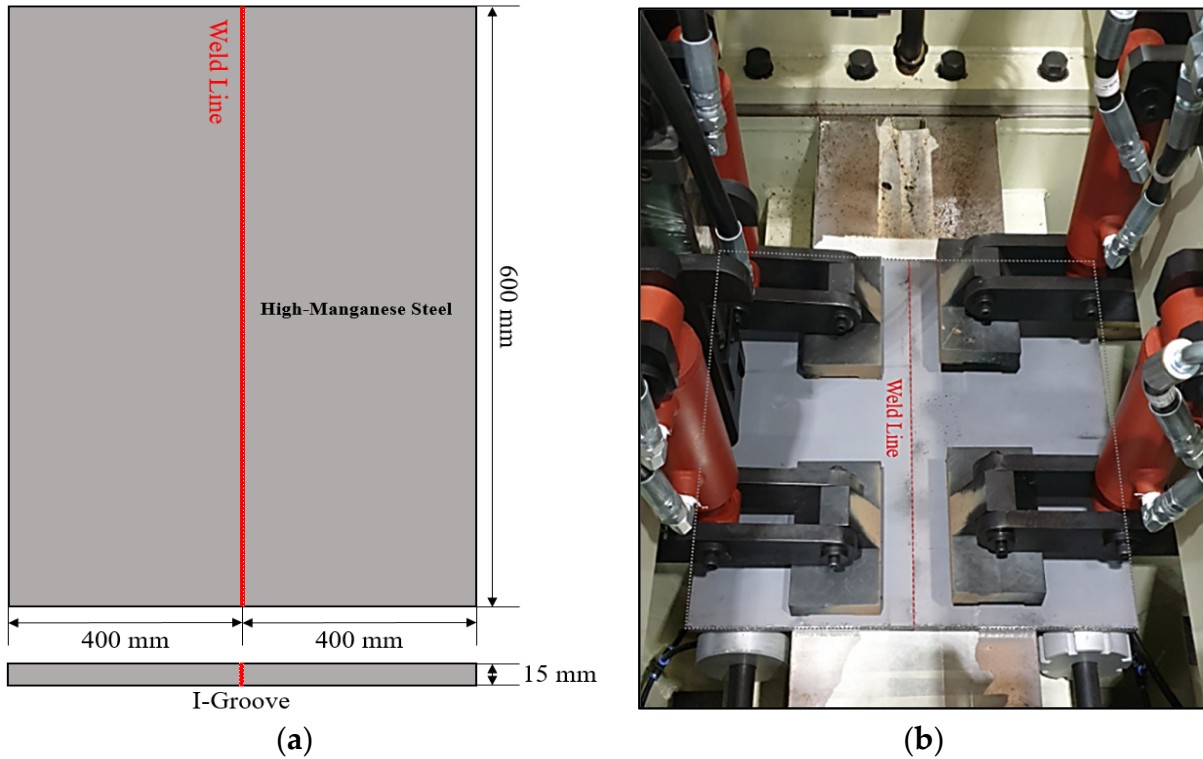

**Figure 1.** Specifications of the test specimen and the experimental setup of the test specimen. (**a**) Dimension of the test specimen, (**b**) experimental setup of the test specimen.

The results of the basic experiment based on the 16 kW laser hybrid system were used to establish optimal welding conditions for the main experiment. Both experiments used the same weld material. Argon (purity 99.997%, Type I, G-11.1 Grade C) was used as the shielding gas and was supplied at rates of 15 and 25 L/min to the weld face and root surface, respectively. A fume collector was installed on the laser head component in the 20 kW laser-MIG hybrid welding system to remove the fumes generated during the welding process, separating them from the shielding gas. The configuration and parameters of the laser-MIG hybrid welding system are shown in Figure 2.

The weld joint obtained under optimized experimental conditions was subjected to a tensile test (according to ASTM E8/E8M-16a) for verifying its ability to withstand a load greater than the maximum strength of the base material. In addition, the area at fracture was examined. Further, impact, bend, and hardness tests were conducted.

## 2.2. Bead-on-Plate (BOP) Test

The laser welding characteristics of high-Mn steel were studied by performing a bead-on-plate (BOP) test prior to the experiment. The penetration depth of each output while performing a single laser welding process with a 16 kW disk laser welding system was examined. The penetration depths of the laser outputs were compared by conducting an experiment using SS400 steel. A welding speed of 100 cm/min (16.7 mm/s) and a laser focal depth (Fd) of 0 mm were used to determine the depth of each laser output.

The penetration depth was observed to increase linearly with laser output. However, for surface beads, the rise observed in high-manganese steel was more significant than that in SS400 steel.

The depth of penetration and the width of the upper bead are compared for each output (kW) for a single laser welding process in Figure 3.

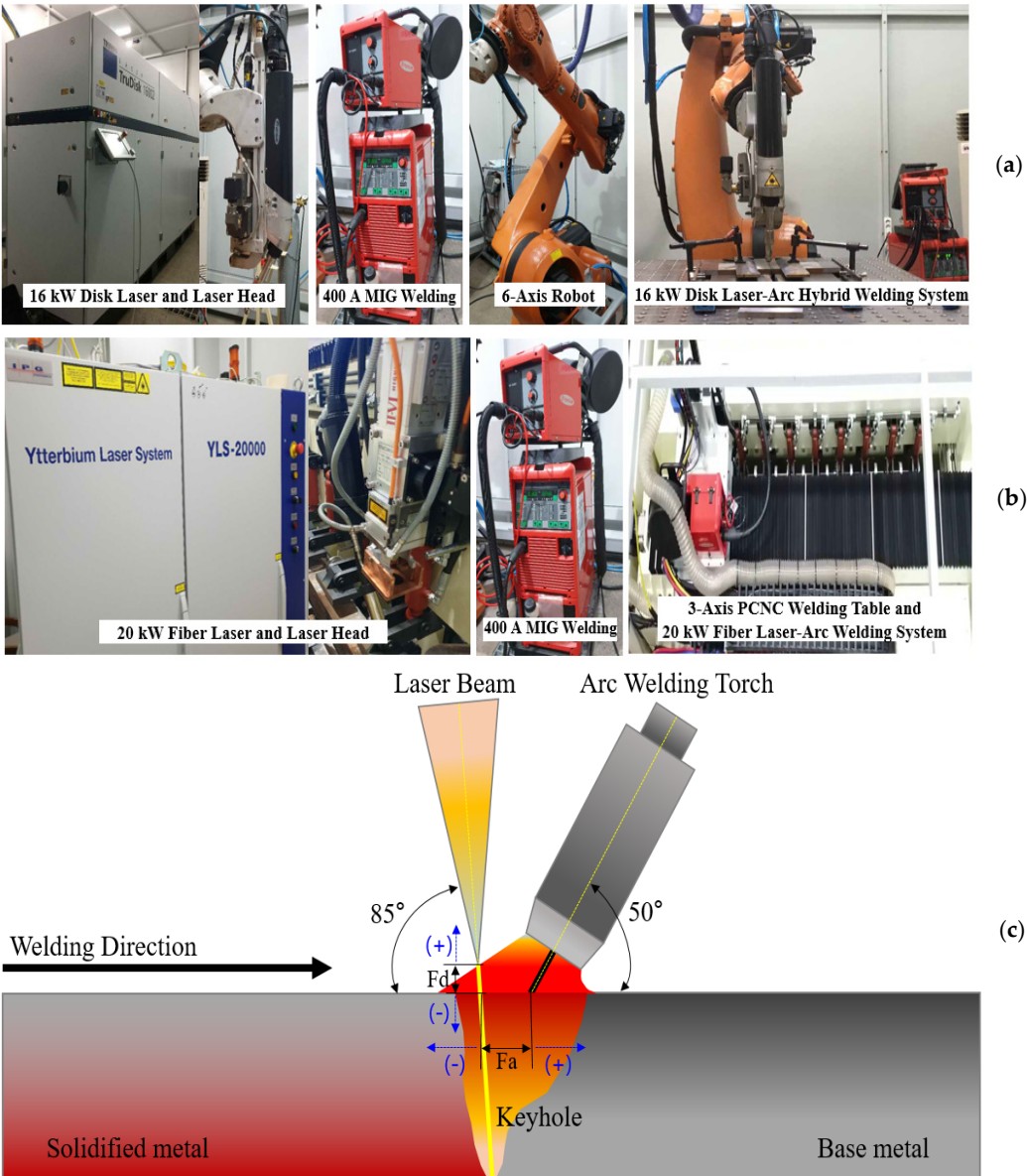

**Figure 2.** Laser-MIG hybrid welding system configurations and a schematic of laser-MIG hybrid welding. (**a**) 16 kW disk laser-MIG hybrid welding system, (**b**) 20 kW fiber laser-MIG hybrid welding system, (**c**) schematic of laser-MIG hybrid welding. Laser focal depth (Fd): distance between the laser focus and the surface of the specimen. Laser-arc distance (Fa): distance between the center of focused laser beam and the tip of the wire.

The difference between the penetration depth values obtained through the laser-arc distance (Fa) and the arc output was examined. The experimental conditions involved a welding speed of 100 cm/min (6.7 mm/s), Fd = −4 mm, and Fa = 4 and 6 mm. The welding current was varied from 150 to 400 A.

As a result of the experiment, when the laser focal depth (Fd) was fixed at −4 mm, the penetration depth did not increase linearly as the arc power changed. This is because the densities of the laser beams that are encapsulated vary according to the focal depth of lasers (Fd). Further, the difference between the width and height of the weld beam formed on the surface is correlated if the laser-arc distance (Fa) is large. In addition, the number of spatters on the weld surface indicates that the focal depth of the beam is inappropriate and needs to be reduced.

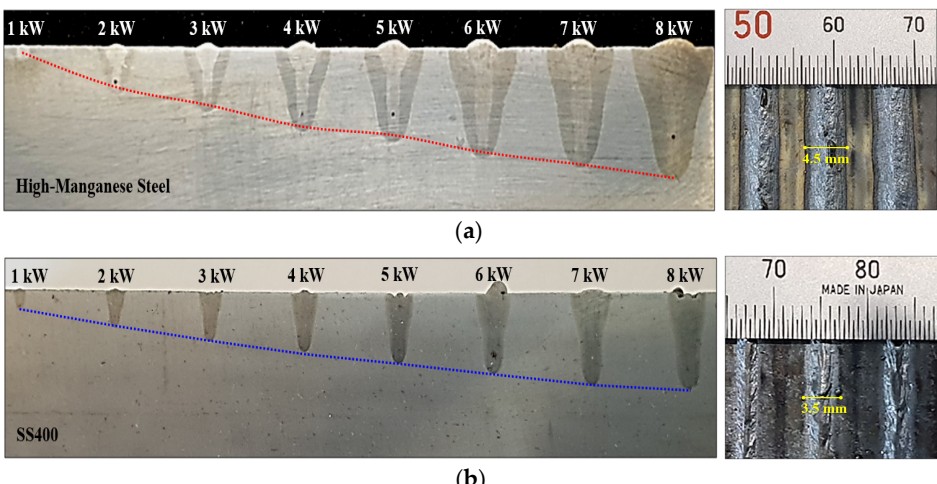

**Figure 3.** Comparisons between the penetration depths and weld face widths of single laser welding outputs for different materials. (**a**) Penetration depth and weld face width of the laser output for high-Mn steel and (**b**) penetration depth and weld face width of the laser output for SS400 steel.

Figure 4 shows the variation of the penetration depth with the distance between the laser and the arc (Fa) during laser-MIG hybrid welding and the spatter aspect around the weld face for Fd of −4 mm.

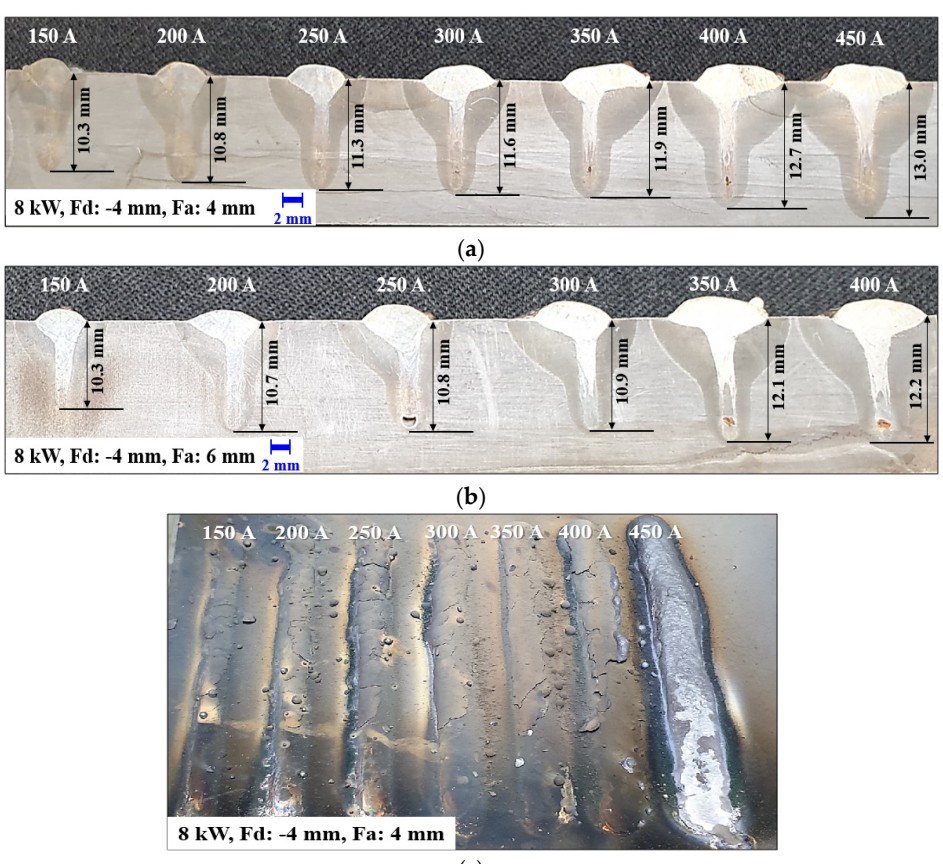

**Figure 4.** Penetration depth and weld face spatter aspects observed during laser-MIG hybrid welding conditions of high-Mn steel. (**a**) Macro cross section of the specimen for a power output of 8 kW, Fd = −4 mm, and Fa = 4 mm; (**b**) macro cross section of the specimen for a power output of 8 kW, Fd = −4 mm, and Fa = 6 mm; (**c**) weld face spatter aspect after the BOP test.

## 3. Results

### 3.1. Analysis of the Main Experiment for Obtaining Optimal Conditions

The main experiment was performed using a 20 kW fiber laser-MIG hybrid welding system according to the results of the previous 16 kW disk laser-MIG hybrid welding system. The basic experiment indicated that a minimum laser power of 12 kW could produce a penetration of 15 mm at a welding speed of 100 cm/min (16.5 mm/s). However, the basic experiment, which was performed with a power output of 8 kW and an Fd = −4 mm, produced a large amount of spatter. Thus, it became necessary to reduce the focal depth (Fd) to negative direction. A slight adjustment in the negative direction was required, in comparison with the results of BOP tests, to reduce the focal size at any wavelength according to the calculated value of the beam focusing process [20,21].

The first test was conducted by shifting the focal depth (Fd) in the negative direction to reduce the laser beam diameter (Test-01). However, a severe humping bead was observed on the root surface of the specimen. The result of Test-01 revealed that the laser beam diameter was insufficient to generate a humping bead. Table 4 shows the humping produced in each experiment for varying experimental conditions.

**Table 4.** Variation of the humping defect shape with the experimental conditions.

| Test No. | Welding Conditions | | | | | Humping Defect | Root Surface Appearance |
| | Laser (kW) | Fa (mm) | Fd (mm) | Welding Current (A) | Speed (cm/min) | | |
|---|---|---|---|---|---|---|---|
| 01 | 13 | 6 | −14 | 300 | 100 | Poor |  |
| 02 | 13 | 6 | −14 | 200 | 100 | Bad |  |
| 03 | 13 | 6 | −14 | 250 | 100 | Poor |  |
| 04 | 14 | 6 | −14 | 400 | 100 | Poor |  |
| 05 | 16 | 6 | −14 | 400 | 100 | Poor |  |
| 06 | 16 | 6 | −17 | 200 | 100 | Poor |  |
| 07 | 16 | 6 | −12 | 200 | 100 | Poor |  |
| 08 | 16 | 6 | −10 | 200 | 100 | Bad |  |
| 09 | 16 | 6 | −8 | 200 | 100 | Poor |  |

**Table 4.** *Cont.*

| Test No. | Welding Conditions | | | | | Humping Defect | Root Surface Appearance |
|---|---|---|---|---|---|---|---|
| | Laser (kW) | Fa (mm) | Fd (mm) | Welding Current (A) | Speed (cm/min) | | |
| 10 | 16 | 6 | −10 | 100 | 100 | Poor |  |
| 11 | 17 | 6 | −10 | 150 | 100 | Bad (Underfill) |  |
| 12 | 18 | 6 | −10 | 200 | 100 | Poor (Underfill) |  |
| 13 | 13 | 6 | −12 | 300 | 100 | Poor |  |
| 14 | 13 | 6 | −11 | 300 | 100 | Poor |  |
| 15 | 13 | 6 | −11 | 300 | 110 | Poor |  |
| 16 | 13 | 6 | −11 | 300 | 90 | Poor |  |
| 17 | 13 | 4 | −11 | 300 | 90 | Poor |  |
| 18 | 13 | 4 | −11 | 300 | 95 | Poor |  |
| 19 | 13 | 4 | −11 | 300 | 100 | Bad |  |
| 20 | 13 | 2 | −11 | 300 | 100 | Poor |  |
| 21 | 14 | 8 | −8 | 300 | 100 | Bad |  |
| 22 | 14 | 8 | −8 | 300 | 102 | Good |  |
| 23 | 14 | 8 | −8 | 300 | 105 | Bad |  |

The visual evaluation criteria were established according to the extent of the humping defect on the root surface. "Poor" was classified for the simultaneous humping defect and meltdown phenomenon on the root surface. The evaluation criterion of "Bad" was selected as the case where a humping defect occurred but was not noticeable, and this established the criterion for a comparison of different welding conditions. The evaluation criteria of "Good" were established as welding conditions in which a continuous weld bead was formed without any weld defect.

Additional experiments were conducted to analyze the occurrence of the humping defect in Test-01. The experimental results from Test-02 to Test-05 were used to compare the outputs of the laser and arc and determine the degree of occurrence of the humping defect. A comparison between Test-01 and Test-02 demonstrated that the reduction in the arc output under identical conditions decreased the humping. Although the lower bead was formed in Test-02, the degree of humping defect was drastically reduced. Similarly, the degree of humping defect observed in Test-03 was lesser than that in Test-01 due to a reduction in the arc output. The results of Test-04 and Test-05 reported a slight reduction in the size of the humping defect owing to the rise in the laser power under identical experimental conditions. However, the simultaneous occurrence of root surface beads and humping defect is noteworthy.

The results of Test-06 to 09 were compared with those of Test-05 by fixing the laser power 16 kW and reducing the arc power (welding current) to 200 A. The reduction in the humping defect, obtained by reducing the heat input of the arc by half, was compared with the experimental conditions of Test-04 to verify the hypothesis that the degree of humping defect was dependent on the laser focal depth. The experiment wherein the Fd values were varied produced a significant humping defect at an Fd of −17 mm, a relatively low arc output in Test-04. The amount of humping defect did not differ drastically even if the Fd values varied more than in the negative direction; further, when the focal depth values were reduced in the negative direction, the laser beam diameter became very small, and the power density of the laser increased. Thus, the sensitivity of its reaction to the formation of the humping defect increased. However, the degree of humping defect for Fd values of −12 and −10 mm was relatively low, followed by an increase at Fd = −8 mm.

According to the Test-08 results, an Fd of −10 mm for a laser power of 16 kW was appropriate. Thus, additional experiments were conducted according to the test results of Test-08. This experiment was conducted to examine the effect of humping defect on the laser output and the reduction in the input heat by decreasing the existing arc output for an Fd of −10 mm. The heat input obtained from the arc output in Test-10 was half of that in Test-08. However, some humping defect occurred in the middle of the specimen. Test-11 reported the presence of a relatively good lower bead, and an underfill defect occurred in the weld face owing to a high laser output of approximately 17–18 kW. The degree of humping defect in Test-12 was relatively reduced. Welding with a laser output greater than 17 kW and a welding speed of 100 cm/min under identical conditions leads to the occurrence of an underfill defect due to the high power of the laser. However, the occurrence of humping defect due to the variation of laser power did not change significantly due to the welding speed limit of 100 cm/min. The previous experiment confirmed the sensitivity of the underfill defect on the weld face while using a large amount of power from the laser. Therefore, additional experiments were conducted to re-establish the welding conditions using the experimental data obtained at 13 kW.

For Test-14, the Fd was −11 mm, which was 1 mm greater than that for Test-13. However, the humping defect was slightly reduced, occurring at the beginning of the welding process. Test-15 and Test-16 studied the variation of the degree of humping defect with the welding speed under identical conditions. It was observed that a rise in welding speed under identical conditions leads to a slight reduction in humping defect.

Obtaining a solution for a laser power input of 13 kW from previous experiments is difficult. Therefore, the test was conducted by increasing the laser power from 13 to 14 kW. The slight reduction in humping defect upon increasing the laser power under identical conditions was assumed to be a part of previous experimental results. Additionally, the sensitivity to underfill defect was accounted for when the laser power was increased under identical conditions. The laser diameter was further reduced by changing the Fd value to −8 mm, thereby increasing its depth by 1 mm. Thus, humping defect was drastically reduced, which was similar to the observations made in Test-21.

According to the results of Test-21, an additional experiment was conducted to reduce humping defect by increasing the welding speed under identical conditions during the previous experiment. The additional experiment (Test-22) revealed that the best weld bead without the presence of humping defect was formed at a welding speed of 102 cm/min (17.0 mm/s). A small increment in welding speed to 105 cm/min (17.5 mm/s) led to the occurrence of a small humping defect (Test-23).

The results of the final experiment revealed that the optimal conditions included a Fa of 8 mm, an Fd of −8 mm, a welding current of 300 A, and a welding speed of 102 cm/min (17.0 mm/s) at 14 kW. The weld face and root surface bead width of the specimen were measured at 8 and 2 mm, respectively. Table 5 shows the shape of the welded joint appearance under optimal welding conditions.

**Table 5.** Optimal welding conditions and welded joint appearance.

| Test No. | Optimal Welding Conditions | | | | | Welded Joint Appearance |
| --- | --- | --- | --- | --- | --- | --- |
| | Laser (kW) | Fa (mm) | Fd (mm) | Welding Current (A) | Speed (cm/min) | |
| 22 | 14 | 8 | −8 | 300 | 102 | 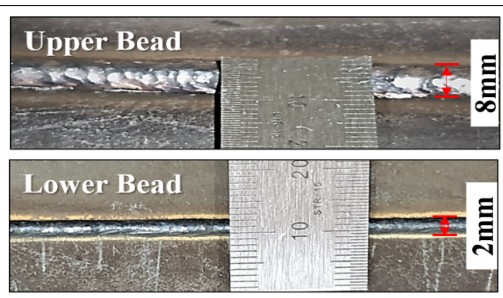 |

### 3.2. Mechanical and Microstructural Characteristics

3.2.1. Tensile Strength Test

The test methods and procedures for the tensile test were performed through the standard of ASTM E8/E8M-16a. The tests were performed using a 100 ton Walter+Bai AG/TTM-1000 (Löhningen, Switzerland) universal testing machine.

As a result of the tensile test, fracture occurred in the weld metal area, but the result was slightly less than the tensile strength of the base material (866 MPa). However, there was no problem because the IGC Code had a value of more than 660 MPa of acceptance criteria.

The yield strength was 30% higher (513.3 MPa) than that of the base material (467 MPa). Figure 5 shows the results of the tensile test and the appearance of the specimen fracture.

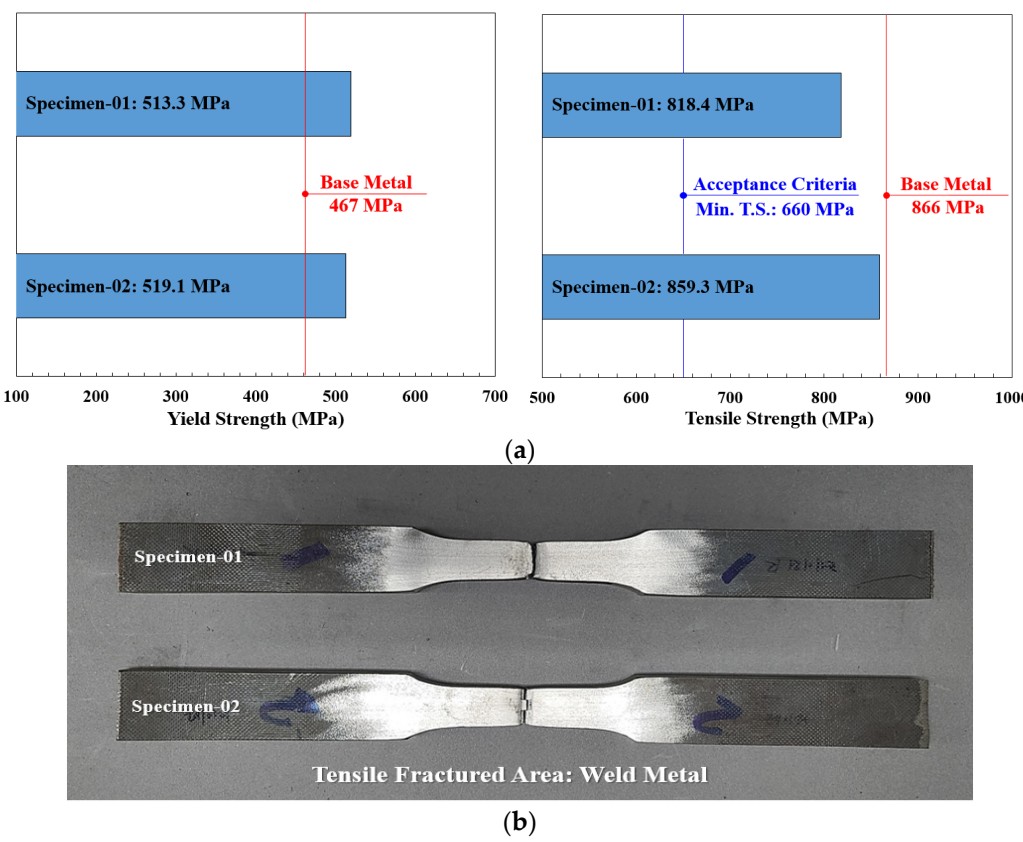

**Figure 5.** Results of the tensile test and the shape of the fracture specimen. (**a**) Yield-tensile strength of the laser-MIG hybrid welded joints, (**b**) shape of the fracture specimen after the tensile test.

### 3.2.2. Low-Temperature Impact Test

Low-temperature impact tests must be conducted on the material specimen to test the viability of using the material for constructing LNG fuel tanks. The IGC Code states that the test should require more than 27 J of energy for a weld metal at $-196\,°C$. The low-temperature impact tests were performed according to the guidelines specified by ASTM E23-16b. The tests were performed using a Tinius Olsen/201892 (Horsham, PA, USA) impact testing machine.

An impact value greater than or equal to 58 J was obtained for each location of the specimen through the experiment. Table 6 lists the low-temperature impact values of each specimen location. The fractured surface is shown in Figure 6.

**Table 6.** Low-temperature impact values for different specimen locations.

| Position | Temperature (°C) | Test Results (J) | Accept Criteria |
|---|---|---|---|
| Weld metal | | 58 | |
| Fusion line | | 62 | $\geq$27 J at $-19\,°C$ at weld metal base |
| Fusion line + 1 mm | $-196$ | 114 | |
| Fusion line + 3 mm | | 112 | |
| Fusion line + 5 mm | | 94 | |

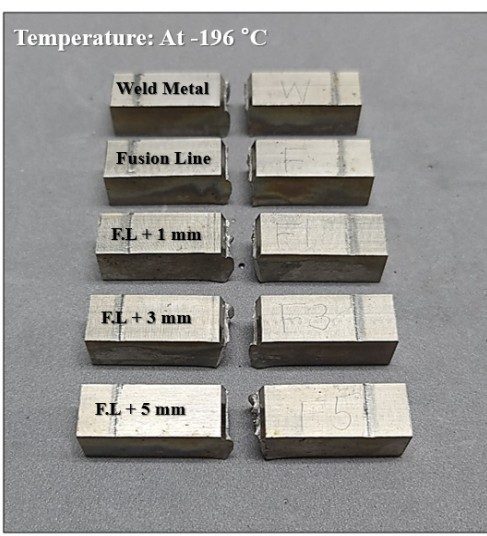

**Figure 6.** Fractured specimen after the low-temperature impact test.

### 3.2.3. Bend Test

A bend test of the welded joint was conducted to visually examine the degree of plastic deformation of the welded joint. The experiment was carried out according to the ASTM E190-14 standard. The tests were performed using a 20-ton R&B/R&B-301 (Daejeon, Korea) impact testing machine.

The results of this test confirmed that the weld joint was completely ductile, even after the plastic deformation. There was no occurrence of a brittle fracture or a crack, and the experimental results indicated a lack of discontinuities in all directions. Figure 7 shows the bending behavior at each position following the bend test.

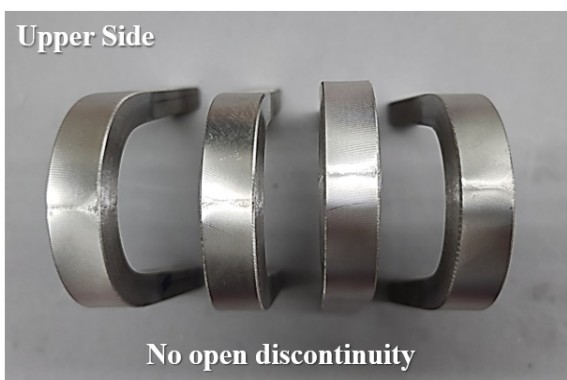

**Figure 7.** Appearance of the specimen after the bend test.

### 3.2.4. Macrostructure Observation and Hardness Test

As laser welding involves rapid cooling of the specimen after welding, there is a sudden change in the temperature of the welded part, which includes the heat-affected region. This could lead to a change in metal structure. Thus, a macro test was performed to study this hardened structure. This was followed by a hardness test, which was conducted on the macrostructure specimen according to the ASTM E92-17 standard. The tests were performed using an AKASHI/HV-113 (Kawasaki, Japan) Vickers hardness test machine.

The hardness test confirmed that the maximum hardness value did not exceed 292 HV. A hardened structure was not observed in the laser-MIG hybrid welding joints of high-Mn steel. In addition, the macrostructure formation revealed the characteristics of a general laser-welded section that was narrow and long and confirmed the absence of internal defects. The sectional macrostructure and the results of the hardness test are shown in Figure 8.

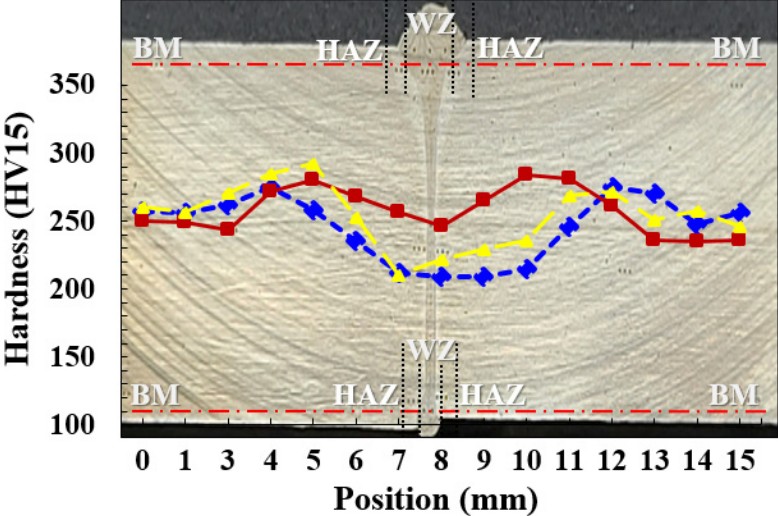

**Figure 8.** Sectional macrostructure of the welded joints and the results of the hardness test.

### 3.2.5. Microstructure of Laser-MIG Hybrid Welded Joint

The microstructure of the cross section of the laser-arc hybrid weld was examined at a magnification of 50× to 200× using an optical microscope under optimal conditions. The cross-sectional microstructure is shown in Figure 9. The laser-MIG hybrid weld was divided into the arc and laser zones for comparison purposes [22].

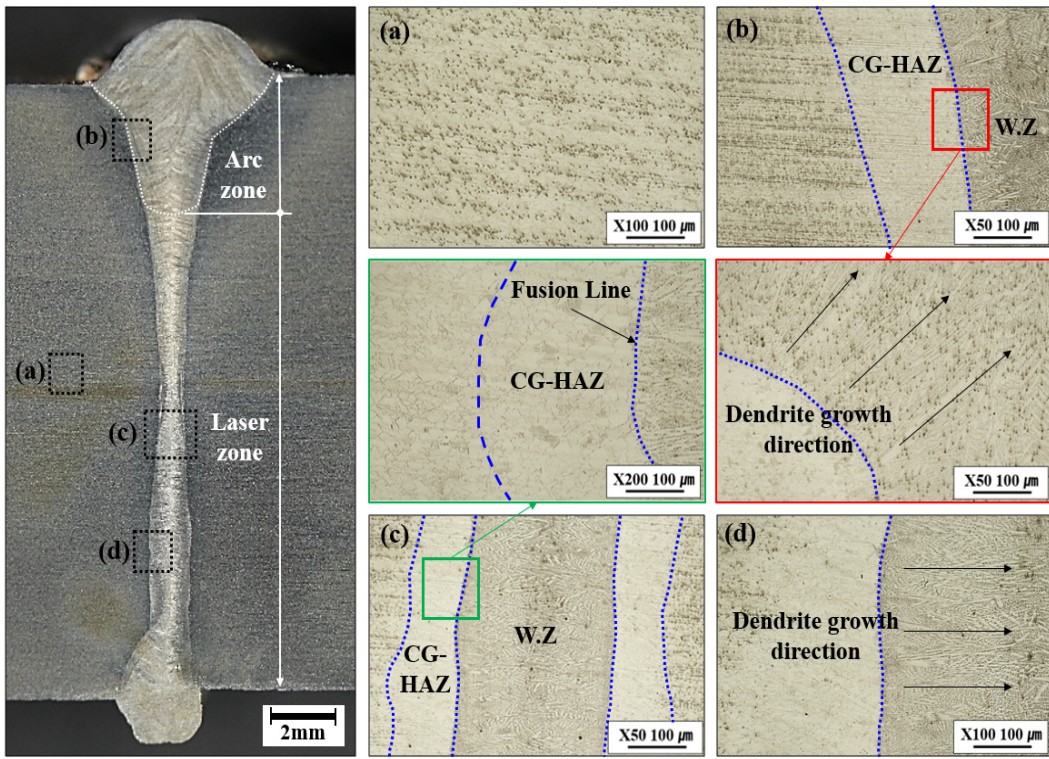

**Figure 9.** Optical micrograph of the laser-MIG hybrid welded joints. (**a**) Base metal, (**b**) WZ + CG-HAZ of the arc zone, (**c**) WZ + CG-HAZ of the laser zone, and (**d**) dendritic growth direction of the laser zone.

The overall laser-MIG hybrid weld did not appear to have defects, such as cracks and pores. In addition, there were no liquation cracks in the fusion boundary and the coarse grain heat-affected zone (CG-HAZ). The columnar dendritic structures in Figure 9b,d propagated from the center of the weld bead to both sides of the base metal as the specimen cooled down [23]. The columnar dendrites grew parallel to the maximum heat flow direction and were aligned with the welding heat source.

The laser zone grew horizontally in the direction of the welding center and along the fusion line because the molten pool became narrower than the arc area. As a result, the rapid growth of the dendritic structure was enhanced with the increase in the cooling rate. Figure 9b,c shows that the CG-HAZ was wider than the laser zone. The differences in the microstructures of the laser hybrid welds at different locations are attributed to the differences in the cooling rate and heat input [24,25].

## 4. Conclusions

This study aimed to determine optimal conditions for performing laser-MIG hybrid welding on a 15 mm thick high-Mn steel specimen at welding speeds greater than 100 cm/min. However, significant humping defect was observed in the root surface of the specimen during the experiment. The main conclusions derived from these experiments are summarized below.

1. Reduction of the focal size of the laser beam by the Fd value in the negative direction instead of on the object surface increased its sensitivity to the occurrence of humping defect. It was also concluded that reducing the laser beam diameter might reduce the underfill defect when using a high-power laser output.
2. The relationship between humping defect and the distance of the laser and arc (Fa) was not very conclusive. A reduction in the arc output identical welding conditions and a high welding speed might reduce the humping defect.
3. Although the tensile strength of the base material was slightly less than 866 MPa under optimal welding conditions, the yield strength was 30% higher than that of the base material. In addition, the low-temperature impact values of the specimen were equal to or greater than 58 J at all locations in the weld zone, which was twice that of the accepted value.
4. Discontinuities and defects were not observed in any direction during the bend test. The hardness test confirmed that the hardness did not exceed 292 HV. Thus, a separately hardened structure was not formed in the welded joint.
5. The microstructure of the weld joint confirmed the absence of defects, such as cracks and pores, in the overall laser-arc hybrid welded joint. In addition, liquation cracks were absent in the fusion boundary and at the CG-HAZ. The differences can be observed clearly by segregating the MIG and laser zones according to the characteristics of the laser-arc hybrid welding section. These differences in the microstructures of the laser hybrid welds at different locations are attributed to the differences in the cooling rate and heat input.

**Author Contributions:** Conceptualization, D.-S.K. and K.-H.L.; data curation, D.-S.K. and K.-H.L.; investigation, D.-S.K., H.-K.L., W.-J.S., K.-H.L. and H.-S.B.; methodology, D.-S.K. and K.-H.L.; supervision, H.-K.L. and H.-S.B.; validation, D.-S.K., K.-H.L. and H.-S.B.; visualization, D.-S.K.; writing—original draft, D.-S.K.; writing—review and editing, D.-S.K., H.-K.L., W.-J.S. and H.-S.B. All authors have read and agreed to the published version of the manuscript.

**Funding:** This research received no external funding.

**Acknowledgments:** The authors acknowledge K.-H.Yun in Daewoo Shipbuilding Marine Engineering (DSME) Welding Engineering R&D Department for their administrative support and experimental preparation.

**Conflicts of Interest:** The authors declare no conflict of interest.

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
