# Peer review of "Experimental Study on Laser-MIG Hybrid Welding of Thick High-Mn Steel Plate for Cryogenic Tank Production"

_jmse, doi:10.3390/jmse9060604_

Round 1
Reviewer 1 Report
Interesting experimental study on the application of Laser-Arc Hybrid Welding, well explained and illustrated.
Suggestion for Authors...I am not sure is there any methodology behind experimentation (any of Design of Experiments method), I did not see it. If there is such a method, please give reference to it. If there is no such "well-known method" please explain what was the changing experiments parameters procedures.
How we (authors) are sure that Weld-22 is the best possible (optimal) weld?
Figure 2, a) and b)...please check arc welding power source caption (is it 40A or 400A?)
Figure 2, c)...short explanation (kind of legend) would be OK for Fd and Fa. They are mentioned latter in the text, but at this point we do not know exactly what these symbols represent.
Author Response
- Suggestion for Authors...I am not sure is there any methodology behind experimentation (any of Design of Experiments method), I did not see it. If there is such a method, please give reference to it. If there is no such "well-known method" please explain what was the changing experiments parameters procedures.
- Authors Response: In general, laser-arc hybrid welding is based on arc leading and laser following processes.[Line111-112] This is commonly used as a general condition for laser-arc hybrid welding. In addition, the parameters for establishing the optimum welding conditions were performed through changes in laser output, Fa, Fd, current, and welding speed, which are described in experimental conditions for each condition.[Table 4]
- How we (authors) are sure that Weld-22 is the best possible (optimal) weld?
- Authors Response: We inspected the integrity of the welds through the visual test and added the visual test criteria. Please refer to revised manuscript.[Line212-218]
- Figure 2, a) and b)...please check arc welding power source caption (is it 40A or 400A?)
- Authors Response: Confirmed, we modified welding power source caption. Please refer to revised manuscript.[Figure 2, a), b)]
- Figure 2, c)...short explanation (kind of legend) would be OK for Fd and Fa. They are mentioned latter in the text, but at this point we do not know exactly what these symbols represent.
- Authors Response: Agreed, we added the meaning of symbol. Please refer to revised manuscript.[Figure 2, c)]
Reviewer 2 Report
In this paper a performance evaluation was conducted according to the weld quality standards specified in the international gas carrier code. The results of this study indicate that it may be possible to use laser-arc hybrid welding on thick high manganese steel plates.
The presented article is well written.
I understand that it is original. However, it seems to me that you can add a few more references, e.g. from the elsevier or scopus database.
Author Response
- In this paper a performance evaluation was conducted according to the weld quality standards specified in the international gas carrier code. The results of this study indicate that it may be possible to use laser-arc hybrid welding on thick high manganese steel plates. The presented article is well written
- Authors Response: Thanks so much for your positive feedback.
- I understand that it is original. However, it seems to me that you can add a few more references, e.g. from the elsevier or scopus database.
- Authors Response: Agreed, we added the reference considering your feedback. Please refer to revised manuscript.[Reference]
Reviewer 3 Report
Dear Authors,
Your article titled: “Experimental Study on Laser-Arc Hybrid Welding of Thick High Manganese Steel Plate for Cryogenic Tank Production” is aimed at assessing weldability of high-alloy manganese steel with austenitic structure intended for cryogenic applications subjected to hybrid welding. I believe that the topic is of great practical importance and work made a positive impression on me. I believe that the subject and content of the article justifies its publication, but some changes should be made. Below I present my remarks and comments in the order they appear in the text.
Please pay attention to the use of the correct welding nomenclature: "welded joint", "face", "root". It is necessary to change the terms: "single pass of the front side", "weld material", "the upper and lower parts", "lower bead appearance ".
In addition, the redundant words "thus" and "however" are often used in the manuscript. Removing them will not change the meaning of the sentences.
In the title, abstract and keywords section, there is no information about which of the hybrid processes was used. Please add the information that this is a laser-MIG.
Abstract needs rewording:
My proposition:
“The International Maritime Organization has recently updated the ship emission standards to reduce atmospheric contamination. One technique for reducing emissions involves the application of liquefied natural gas (LNG). LNG is an eco-friendly ship fuel. However, the stable storage and transport of LNG can be realized by reducing its volume to 1/600 of its original value by super-freezing and liquefying it at temperatures equal to or below −163℃. Thus, the tanks used for the transport and storage of LNG, must have very low thermal expansion capability and high cryogenic toughness. For excellent cryogenic properties, high-manganese steel with a complete austenitic structure, is used to design these tanks. This study aims to determine, the optimum welding conditions for performing laser-arc hybrid welding through the arc leading and laser following processes. A welding speed of 100 centimeter per minute cm/min was used for welding a 15 mm (t) thick high-manganese steel plate. A performance evaluation was conducted according to the weld quality standards specified in the international gas carrier code. The quality of the butt weld was found to be satisfactory. The results of this study indicate that it may be possible to use laser-arc hybrid welding on thick high manganese steel plates.”
Please add information about the name of the process used, list the tests that the welded joints were subjected to and add the results of these tests in quantitative form.
Introduction:
The phrase "encourages ships"… doesn't sound right, too anthropomorphic. Maybe it is better to write that shipowners are encouraged?
The section is not too long, it would be beneficial to extend it with information about laser, MIG and hybrid processes: scope of application, advantages, disadvantages. This will allow you to simultaneously increase the number of publications in the references section (see last remark).
In this regard, I can recommend the following current scientific articles (not mine):
https://doi.org/10.3390/ma13204540, 10.2478/adms-2019-0002
There is no need to write the names of chemical elements with a capital letter and the translation of their chemical symbols (line 52).
Line 42: Please correct the grammar of this sentence.
Line 56: There are no references to the SAW and FCAW processes. Please consider e.g. article: https://doi.org/10.1016/j.conbuildmat.2019.117697;
Please use SI units: cm (line 87)
Materials and Methods:
Please enter process name: laser-MIG
Table 2: please correct the notation of units: MPa.
Line 100 and table 3: change "weld material" to: "consumable"
According to the Mdpi guidelines, all devices used in the tests must be properly described.
Line 120: what grade of Ar was used (designation)?
There is no information that VT studies have been conducted.
Table 4: change "Arc" to: "welding current". Correct the notation: “underfill”.
Chapter 3.2.1: the first paragraph should be placed within the MandM chapter.
Figure 5: change Mpa to MPa and add space before unit. Figure caption: correct font size.
Figure 6: One of the photos is not needed.
Figure 7: The top two photos are not needed. If they are really valid, all photos should be labeled (a), (b), (c).
Chapter 3.2.4 title: "Macro" doesn't sound technical.
Figure 8 caption: correct the notation: "macrostructure".
Chapter 3.2.4 title: what does "Microstructure of characteristics" mean?
Line 348: remove: "weld"
Chepter 4 should be titled: “conclusions”.
There are no mandatory parts in the manuscript: Authors contributions e.t.c.
References section contains too few articles. A scientific article with such content should, in my opinion, have over 25 recent articles as a references.
Author Response
- Dear Authors, Your article titled: “Experimental Study on Laser-Arc Hybrid Welding of Thick High Manganese Steel Plate for Cryogenic Tank Production” is aimed at assessing weldability of high-alloy manganese steel with austenitic structure intended for cryogenic applications subjected to hybrid welding. I believe that the topic is of great practical importance and work made a positive impression on me. I believe that the subject and content of the article justifies its publication, but some changes should be made. Below I present my remarks and comments in the order they appear in the text.
- Authors Response: Thanks so much for your positive feedback.
- Please pay attention to the use of the correct welding nomenclature: "welded joint", "face", "root". It is necessary to change the terms: "single pass of the front side", "weld material", "the upper and lower parts", "lower bead appearance ".
- Authors Response: Agreed, we corrected as the misnomers. Please refer to revised manuscript.
- In addition, the redundant words "thus" and "however" are often used in the manuscript. Removing them will not change the meaning of the sentences.
- Authors Response: Thanks so much for your helpful feedback.
- In the title, abstract and keywords section, there is no information about which of the hybrid processes was used. Please add the information that this is a laser-MIG.
- Authors Response: Agreed, we corrected the laser hybrid welding process. Please refer to revised manuscript.
- Abstract needs rewording:
My proposition:
“The International Maritime Organization has recently updated the ship emission standards to reduce atmospheric contamination. One technique for reducing emissions involves the application of liquefied natural gas (LNG). LNG is an eco-friendly ship fuel. However, the stable storage and transport of LNG can be realized by reducing its volume to 1/600 of its original value by super-freezing and liquefying it at temperatures equal to or below −163℃. Thus, the tanks used for the transport and storage of LNG, must have very low thermal expansion capability and high cryogenic toughness. For excellent cryogenic properties, high-manganese steel with a complete austenitic structure, is used to design these tanks. This study aims to determine, the optimum welding conditions for performing laser-arc hybrid welding through the arc leading and laser following processes. A welding speed of 100 centimeter per minute cm/min was used for welding a 15 mm (t) thick high-manganese steel plate. A performance evaluation was conducted according to the weld quality standards specified in the international gas carrier code. The quality of the butt weld was found to be satisfactory. The results of this study indicate that it may be possible to use laser-arc hybrid welding on thick high manganese steel plates.”
- Authors Response: Agreed, we modified the abstract as your feedback. Please refer to revised manuscript.[Abstract]
- Please add information about the name of the process used, list the tests that the welded joints were subjected to and add the results of these tests in quantitative form.
- Authors Response: Agreed, we modified the abstrct as your feedback. Please refer to revised manuscript.[Abstract]
- Introduction: The phrase "encourages ships"… doesn't sound right, too anthropomorphic. Maybe it is better to write that shipowners are encouraged?
- Authors Response: Agreed, we modified the introduction as your feedback. Please refer to revised manuscript.[Line32-33]
- The section is not too long, it would be beneficial to extend it with information about laser, MIG and hybrid processes: scope of application, advantages, disadvantages. This will allow you to simultaneously increase the number of publications in the references section (see last remark). In this regard, I can recommend the following current scientific articles (not mine): https://doi.org/10.3390/ma13204540, 10.2478/adms-2019-0002
- Authors Response: Agreed, we added the reference related laser-MIG hybrid welding process. Please refer to revised manuscript.[Reference]
- There is no need to write the names of chemical elements with a capital letter and the translation of their chemical symbols (line 52).
- Authors Response: Confirmed, we delected the name of chemical elements as your feedback. Please refer to revised manuscript.
- Line 42: Please correct the grammar of this sentence.
- Authors Response: Confirmed, we corrected the grammar of sentence as your feedback. Please refer to revised manuscript.[Line43-44]
- Line 56: There are no references to the SAW and FCAW processes. Please consider e.g. article: https://doi.org/10.1016/j.conbuildmat.2019.117697;
- Authors Response: Agreed, we added the reference related laser-MIG hybrid welding process. Please refer to revised manuscript.[Reference]
- Please use SI units: cm (line 87)
- Authors Response: Confirmed, we corrected to SI units as your feedback. Please refer to revised manuscript.
- Materials and Methods: Please enter process name: laser-MIG
- Authors Response: Confirmed, we added the process name as your feedback. Please refer to revised manuscript.
- Table 2: please correct the notation of units: MPa.
- Authors Response: Confirmed, we corrected the notation of units as your feedback. Please refer to revised manuscript.
- Line 100 and table 3: change "weld material" to: "consumable"
- Authors Response: Confirmed, we corrected the “weld material” to “welding consumable” as your feedback. Please refer to revised manuscript.[Line103, Table 3]
- According to the Mdpi guidelines, all devices used in the tests must be properly described.
- Authors Response: Agreed, we added the information of test devices as your feedback. Please refer to revised manuscript.[Line302-303, 324-325, 344-345, 360-361]
- Line 120: what grade of Ar was used (designation)?
- Authors Response: Confirmed, we added the grade of used shielding gas as your feedback. Please refer to revised manuscript.[Line123]
- There is no information that VT studies have been conducted.
- Authors Response: Agreed, we added the information of visual test studies as your feedback. Please refer to revised manuscript.[Line212-218]
- Table 4: change "Arc" to: "welding current". Correct the notation: “underfill”.
- Authors Response: Confirmed, we corrected the “Arc” to “welding current” and “under_fill” to “underfill” as your feedback. Please refer to revised manuscript.[Table 4]
- Chapter 3.2.1: the first paragraph should be placed within the MandM chapter.
- Authors Response: Agreed, we replaced the first paragraph chaper 3.2.1 within the MandM chapter as your feedback. Please refer to revised manuscript.[Line129-132]
- Figure 5: change Mpa to MPa and add space before unit. Figure caption: correct font size.
- Authors Response: Confiremd, we corrected the mistyping words as your feedback. Please refer to revised manuscript.[Figure 5]
- Figure 6: One of the photos is not needed.
- Authors Response: Agreed, we delected the unnecessary photos as your feedback. Please refer to revised manuscript.[Figure 6]
- Figure 7: The top two photos are not needed. If they are really valid, all photos should be labeled (a), (b), (c).
- Authors Response: Agreed, we delected the unnecessary photos as your feedback. Please refer to revised manuscript.[Figure 7]
- Chapter 3.2.4 title: "Macro" doesn't sound technical.
- Authors Response: Agreed, we corrected the “Macro” to “Macrostructure observation” as your feedback. Please refer to revised manuscript.[Chapter 3.2.4]
- Figure 8 caption: correct the notation: "macrostructure".
- Authors Response: Agreed, we corrected the “macro” to “macrostructure” as your feedback. Please refer to revised manuscript.[Figure 8]
- Chapter 3.2.4 title: what does "Microstructure of characteristics" mean?
- Authors Response: Agreed, we corrected the “Microstructure of characteristics” to “Microstructure of laser-MIG hybrid welded joint” as your feedback. Please refer to revised manuscript.[Figure 8]
- Line 348: remove: "weld"
- Authors Response: Confirmed, we delected the “weld” as your feedback. Please refer to revised manuscript.[Figure 8]
- Chapter 4 should be titled: “conclusions”.
- Authors Response: Agreed, we corrected the title of Chapter 4 as your feedback. Please refer to revised manuscript.[Chapter 4]
- There are no mandatory parts in the manuscript: Authors contributions e.t.c.
- Authors Response: Agreed, we added the authors contributions in the manuscript.[Authors contributions]
- References section contains too few articles. A scientific article with such content should, in my opinion, have over 25 recent articles as a references.
- Authors Response: Agreed, we added the reference considering your feedback. Please refer to revised manuscript.[Reference]
Reviewer 4 Report
The paper “Experimental Study on Laser-Arc Hybrid Welding of Thick 2 High Manganese Steel Plate for Cryogenic Tank Production” aims to determine, the optimum welding conditions for performing laser-arc hybrid welding through the arc leading and laser following processes. The paper is written in a clear way, although the novelty of the paper on the abstract. “may be possible (line 26)” does not sound very promising the images are not in a standardized format along the paper.
Nevertheless, this work is very interesting, the paper is more on the side of a technical experimental report than of a scientific article in which there is a substantive discussion.
Author Response
- The paper “Experimental Study on Laser-Arc Hybrid Welding of Thick 2 High Manganese Steel Plate for Cryogenic Tank Production” aims to determine, the optimum welding conditions for performing laser-arc hybrid welding through the arc leading and laser following processes. The paper is written in a clear way, although the novelty of the paper on the abstract. “may be possible (line 26)” does not sound very promising the images are not in a standardized format along the paper.
- Authors Response: Agreed, we modified the sentence as your comment. Please refer to revised manuscript.[Line27]
- Nevertheless, this work is very interesting, the paper is more on the side of a technical experimental report than of a scientific article in which there is a substantive discussion.
- Authors Response: Thanks so much for your positive feedback.
Round 2
Reviewer 3 Report
Dear Authors,
thank you for all the answers and for including my comments and suggestions in your manuscript. During the author's proofreading, pay attention to the bibliographic description of reference [21]: It should be: "Landowski M." and "Advances in Materials Science".
Best regards,
reviewer
Reviewer 4 Report
Any comment.